# A printable hydrogel microarray for drug screening avoids false positives associated with promiscuous aggregating inhibitors

Rabia Mateen[1], M. Monsur Ali[2] & Todd Hoare[1,3]

A significant problem in high-throughput drug screening is the disproportionate number of false hits associated with drug candidates that form colloidal aggregates. Such molecules, referred to as promiscuous inhibitors, nonspecifically inhibit multiple enzymes and are thus not useful as potential drugs. Here, we report a printable hydrogel-based drug-screening platform capable of non-ambiguously differentiating true enzyme inhibitors from promiscuous aggregating inhibitors, critical for accelerating the drug discovery process. The printed hydrogels can both immobilize as well as support the activity of entrapped enzymes against drying or treatment with a protease or chemical denaturant. Furthermore, the printed hydrogel can be applied in a high-throughput microarray-based screening platform (consistent with current practice) to rapidly ( <25 min) and inexpensively identify only clinically promising lead compounds with true inhibitory potential as well as to accurately quantify the dose–response relationships of those inhibitors, all while using 95% less sample than required for a solution assay.

[1] School of Biomedical Engineering, McMaster University, Hamilton, ON L8S 4K1, Canada. [2] Biointerfaces Institute, McMaster University, Hamilton, ON L8S 4L8, Canada. [3] Department of Chemical Engineering, McMaster University, Hamilton, ON L8S 4L7, Canada. Correspondence and requests for materials should be addressed to T.H. (email: hoaretr@mcmaster.ca)

High-throughput screening approaches for identifying lead compounds have been widely and successfully used to accelerate the drug discovery process[1]. Optimizing drug lead identification is particularly important when it comes to pressing clinical issues such as antibiotic resistance due to the β-lactamase (β-lac)-mediated degradation of β-lactam antibiotics[2], motivating intensive drug discovery activities in identifying new β-lac inhibitors that can reclaim antibiotics that have previously been rendered ineffective[3]. Current techniques for high-throughput drug screens for this and other drug discovery goals are limited by two key factors. First, the typical microplate assays used require significant sample volumes and are thus relatively costly to run, particularly when screening higher value and/or synthetically demanding compounds[4]. Replacing microplates with microarrays in which target proteins are immobilized on a substrate (and thus interactions occur only interfacially) would significantly reduce the required assay volumes while preserving or even enhancing assay sensitivity and specificity relative to solution-based methods[5,6]. However, while a range of protein immobilization methods including enzyme crosslinking[7], physical adsorption[8] or covalent attachment[9] to a support, or physical entrapment in a polymer network[10], silica-based sol–gel[11], or metal organic framework[12] have been explored, all suffer from drawbacks in terms of their reproducibility and capacity for stably immobilizing target proteins, limiting their utility in drug screening[13].

Second, current screening methods suffer from many false-positive hits[14], with compounds that behave nonspecifically (i.e. promiscuously) often incorrectly identified as promising drug candidates during screening; subsequent secondary screening of these false-positives results in time and money invested in lead compound candidates that are not truly functional inhibitors[15]. Such promiscuous inhibition is typically linked to the tendency for some compounds to self-associate and form colloidal aggregates that sterically, rather than biologically, inhibit binding to active sites of a range of structurally and functionally unrelated enzymes[16]. The inhibitory interaction typically occurs because of protein adsorption onto the surface of aggregates[17], which sequesters enzymes away from their substrates while also often resulting in partial protein denaturation[18]. Significant effort has been invested in examining the nature of these aggregates and determining methods to identify compounds demonstrating aggregative potential[19–21], with only limited success. Because the formation of aggregates can occur over minor changes in concentration, it is difficult to predict potential aggregators strictly based on physical properties[22]. Computational models have been designed to predict the presence of these compounds in pharmaceutical libraries, but have been shown to regularly generate both false-positive and false-negative results[22]. Furthermore, the addition of a non-ionic detergent can disrupt some colloidal aggregates[23], but cannot fully prevent aggregation and has been shown to interfere with other assay components[24], creating challenges with reliable quantification.

In the context of these challenges, hydrogel-based enzyme immobilization platforms offer particular promise. The high water-binding capacity of hydrogels can preserve enzyme hydration over a broad range of storage/application conditions[25–27], promote high enzyme mobility and flexibility[28], and maintain physiologically mimetic conditions for optimal enzyme-catalyzed reactions[29]. In addition, the tunable porosity of hydrogels can enable selective transport of substrates to and from the entrapped enzyme via size selectivity[30], offering potential to sterically block a drug aggregate from reaching the enzyme-binding site and thus minimize (or even eliminate) issues associated with promiscuous inhibition. Interfacial thin-film hydrogels are particularly attractive since they can minimize the kinetic/diffusional drawbacks

associated with bulk hydrogels in biosensing applications, promoting fast assay speeds[31] while maintaining the benefits of size selectivity[32]. Several methods have been developed to fabricate thin-layer interfacial hydrogels on various substrates, including dip-coating[33], spray deposition[34], spin-coating[35], and drop-on demand printing[36]. Printing is particularly advantageous since it is amenable to dispensing small volumes (minimizing sample volumes for screening), can localize materials in specific patterns (enabling, for example, facile printing of multisample arrays on a substrate), and can be scaled to commercial production[37–39].

Herein, we report the fabrication of a printable hydrogel microarray that can both immobilize and stabilize a wide range of enzymes on a cellulose-based substrate and demonstrate the utility of these printed gels in a high-throughput drug-screening application that can discriminate between actual and promiscuous inhibitors of β-lac. We demonstrate that our printable enzyme-immobilizing/stabilizing hydrogel microarrays can enable both quantitative predictions of the $IC_{50}$ values of real inhibitors as well as discriminate true inhibitors from promiscuous aggregating inhibitors within a high-throughput assay format, potentially impactful for more rapidly and inexpensively identifying clinically promising leads.

## Results

**Printing a thin layer hydrogel**. A drop-on-demand syringe solenoid printer was used to sequentially print hydrazide (POH) and aldehyde (POA) functionalized poly(oligoethylene glycol methacrylate) (PO) precursor polymers, previously shown to rapidly gel upon mixing via hydrazone bond formation[40], on a nitrocellulose substrate (Fig. 1a and Supplementary Fig. 1 for $^1$H NMR spectra of polymers). Polymer inks were prepared at 6 w/w % in 10 mM phosphate-buffered saline (PBS), with glycerol added at 5 w/w% to both adjust the viscosity to enhance printability[41] and act as a humectant to avoid nozzle clogging during printing[42]. Gelation was validated by examining whether polymers remained immobilized at their printed positions when exposed to a methanol–water chromatographic separation process. Fluorescently labeled POH (FITC-POH) or POA (Rhodamine-POA) polymers were printed alone, with an unfunctionalized PO polymer (incapable of covalent crosslinking with POH or POA), or with the corresponding unlabeled reactive polymer precursors (POA or POH, respectively; Fig. 1b). When FITC-POH or Rhodamine-POA was printed alone (Fig. 1b, panels i and iv) or with unfunctionalized PO polymer (Fig. 1b, panels ii and v), the fluorescent precursor could transport up the nitrocellulose strip, indicating poor immobilization; conversely, when the reactive POA and POH polymers were sequentially printed in either sequence (Fig. 1b, panels iii and vi), the labeled polymer remained localized at the printed site, suggesting effective gelation. Attenuated total reflectance-Fourier-transform infrared spectroscopy (ATR-FTIR) further confirmed successful deposition of the polymer inks at the paper surface (Supplementary Fig. 2).

Printed samples were subsequently analyzed to investigate the chemical, morphological, and interfacial changes made to the nitrocellulose substrate following hydrogel deposition. X-ray photoelectron spectroscopy (XPS) measurements on samples sequentially printed with POA and POH indicated a peak in the high-resolution nitrogen spectrum at 401.7 eV that corresponds to the −C=N functional group characteristic of a hydrazone bond (Fig. 1c and Supplementary Fig. 3). Scanning electron microscope (SEM) images of vigorously washed gel-printed nitrocellulose strips indicate that the rough and bulbous morphology of unmodified nitrocellulose remains unchanged when (unreactive) PO and POH are sequentially printed, consistent with these polymers being removed from the substrate during the washing

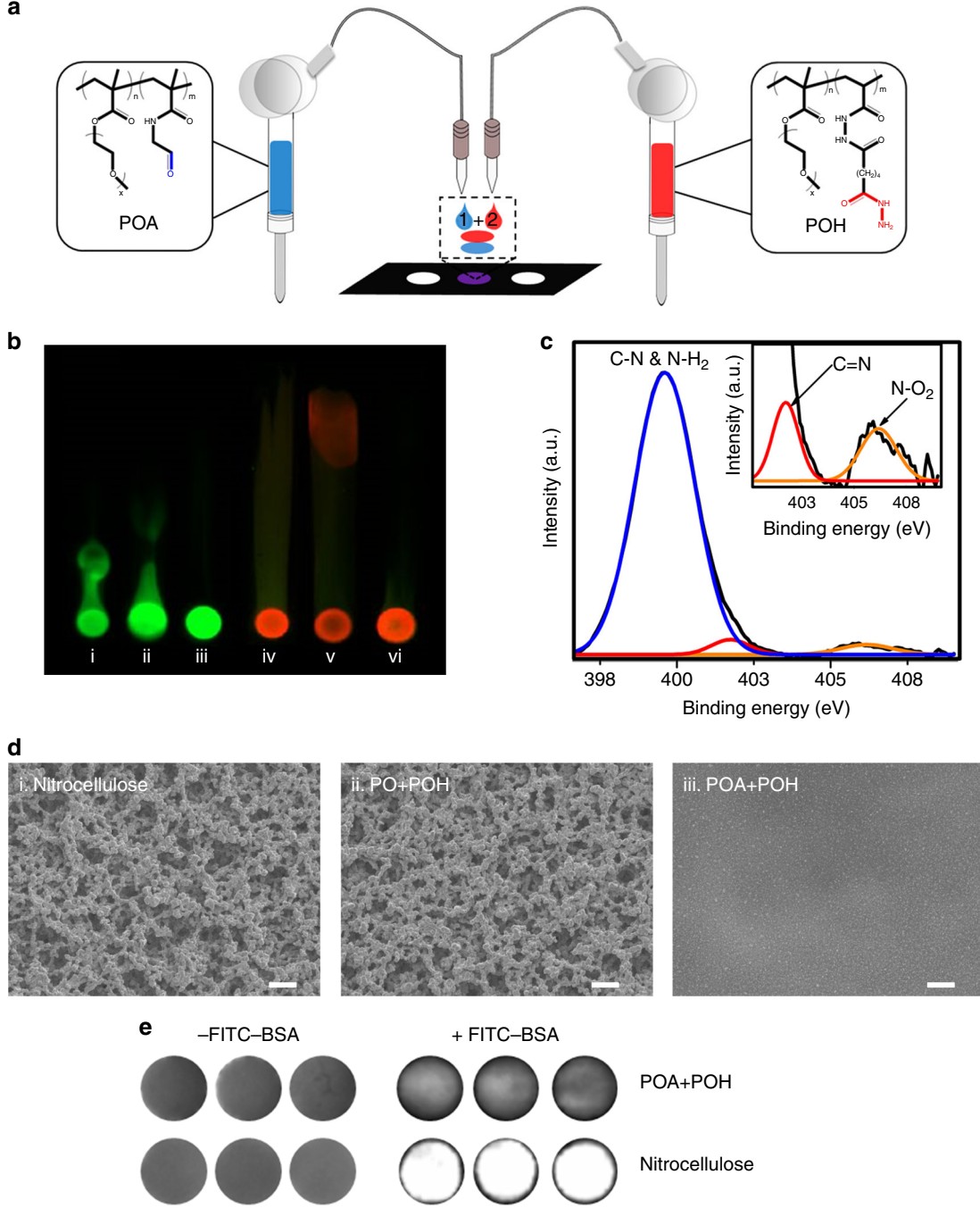

**Fig. 1** Thin layer hydrogels can be printed on nitrocellulose. **a** Schematic of aldehyde-functionalized poly(oligoethylene glycol methacrylate) (POA) and hydrazide-functionalized poly(oligoethylene glycol methacrylate) (POH) polymers sequentially printed onto a nitrocellulose paper substrate using a solenoid-controlled drop-on-demand printing system. **b** Chromatographic separation of printed polymers on nitrocellulose strips in 70:30 methanol: water: (i) fluorescein isothiocyanate-labeled hydrazide-functionalized poly(oligoethylene glycol methacrylate) (FITC-POH) alone; (ii) FITC-POH printed on top of unfunctionalized poly(oligoethylene glycol methacrylate) (PO) (PO+FITC-POH); (iii) FITC-POH printed on top of aldehyde-functionalized poly (oligoethylene glycol methacrylate) (POA) (POA+FITC-POH); (iv) Rhodamine-labeled aldehyde-functionalized poly(oligoethylene glycol methacrylate) (Rhodamine-POA) alone; (v) Rhodamine-POA printed on top of PO (PO+Rhodamine-POA); (vi) Rhodamine-POA printed on top of POH (POH +Rhodamine-POA). **c** High-resolution XPS spectra of printed hydrogel (POA + POH) on a nitrocellulose substrate collected in the N 1 s region. The peak at 401.7 eV corresponds to the –C=N group characteristic of a hydrazone bond, confirming gel formation. **d** SEM images of (i) bare nitrocellulose, (ii) an uncrosslinked polymer assembly (PO+POH) printed on nitrocellulose, and (iii) a printed hydrogel (POA+POH) on nitrocellulose following vigorous washing of the samples in 10 mM PBS. Scale bars, 10 μM. **e** Fluorescence scans of bare nitrocellulose and printed hydrogel (POA+POH) samples before and after incubation in 100 μg ml$^{-1}$ FITC-BSA show a significant reduction in nonspecific protein adsorption following the printing of the hydrogel on the substrate

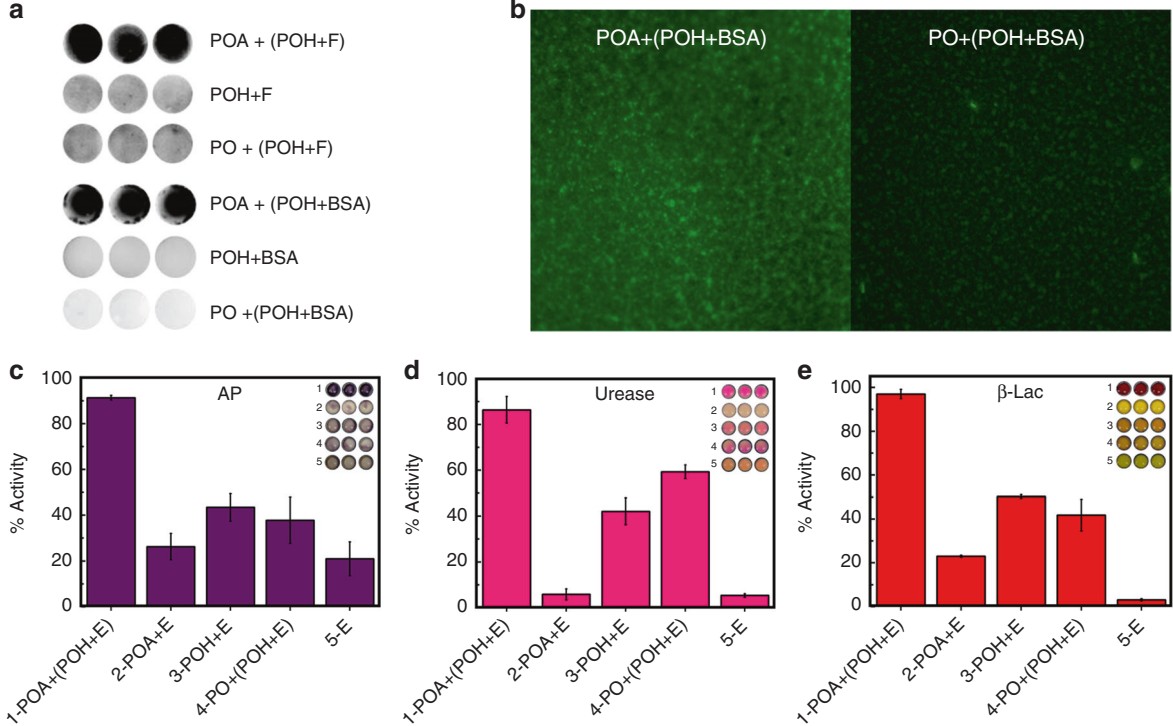

**Fig. 2** Printed hydrogels can immobilize molecules of varying sizes. **a** Retention of fluorescein (F, ~332 Da) printed in hydrogel (POA+(POH+F)) following washing in 0.1 M NaOH + 0.1% Tween 20 and fluorescein isothiocyanate-labeled bovine serum albumin (FITC-BSA, ~66 kDa) printed in hydrogel (POA +(POH+BSA)) following washing in 10 mM PBS for 30 min. The high retained fluorescence signal confirms the efficacy of gel inks for protein immobilization. The three adjacent images in each series are replicates, confirming the reproducibility of the result. **b** FITC-BSA printed in a gelling ink (POA+POH) and a non-gelling ink (PO+POH) imaged by fluorescence microscopy following washing in 10 mM PBS for 30 min (20× magnification). **c–e** Residual activity of enzymes (E) following washing of samples in 10 mM PBS for 10 min, normalized relative to the corresponding unwashed control. **c** Alkaline phosphatase (~69 kDa). **d** Urease (~546 kDa). **e** β-lactamase (~29 kDa). Values are represented as mean ± SD ($n = 3$)

step (Fig. 1d, panels i and ii); conversely, printing the (reactive) POA+POH pair results in significant smoothing of the substrate consistent with the formation of an interfacial gel layer (Fig. 1d, panel iii). The printed hydrogel also significantly suppresses nonspecific protein adsorption to the nitrocellulose substrate (Fig. 1e), a notable benefit for optimizing the sensitivity of any bioassay. This degree of protein repellency is consistent with our previous work on sequential dip-coating of POA/POH on cellulose paper;[33] however, the printing method is both significantly faster (seconds as opposed to hours[33]) and enables localized gel printing essential for creating microarrays. This method allows for a covalently crosslinked hydrogel to be printed in the absence of ultraviolet (UV) photopolymerization to facilitate gelation[43], avoiding the need for a secondary processing step as well as eliminating the potential for enzyme denaturation sometimes observed because of the radical species generated during polymerization[44].

**Entrapping biomolecules in the printed hydrogel.** The printed hydrogel can entrap molecules of varying sizes and effectively immobilize them on the nitrocellulose surface. POH ink solutions were prepared with dissolved fluorescein (332 Da) or FITC-BSA (bovine serum albumin; 66.5 kDa) and subsequently layer-by-layer printed with POA. Both fluorescein (POA+(POH+F)) and BSA (POA+(POH+BSA)) remained entrapped in the crosslinked polymer assembly after the samples were washed vigorously, while the POH+F or POH+BSA ink printed alone or with an unreactive (PO) polymer resulted in almost complete washing of immobilized agent from the surface (Fig. 2a). Printed samples

subjected to chromatographic separation similarly showed minimal transport of the fluorescent dopants from the gel-printed samples but rapid transport when the dopants were printed alone or with an unreactive PO polymer (Supplementary Fig. 4). Fluorescence microscopy images of printed FITC-BSA confirmed the uniform distribution of the protein on the nitrocellulose surface when entrapped in the thin layer hydrogel (Fig. 2b), while confocal microscopy images of FITC-BSA encapsulated inside a hydrogel prepared with Rhodamine-POA confirm that the printed protein is distributed evenly throughout both the cross-section and the depth of the printed hydrogel microzones (Supplementary Fig. 5).

Subsequently, alkaline phosphatase (AP), urease (Ur), and β-Lac were printed alone (E) or in POA or POH ink (POA+E or POH+E) on top of either untreated nitrocellulose or nitrocellulose pre-printed with POA or (unfunctionalized) PO. All tested enzymes were effectively immobilized and stabilized in the printed hydrogel (POA+(POH+E)), with >90% activity maintained for AP and β-Lac and >85% activity maintained for urease relative to enzymes printed in the same manner but not rinsed prior to activity testing (Figs. 2c–e). In comparison, co-printing enzymes with POH alone, POA alone, or in combination with unfunctionalized PO showed only limited benefits in terms of immobilizing and stabilizing the enzymes. Note that, although nitrocellulose has a high capacity for protein retention[45], printed enzyme did not remain associated with unmodified nitrocellulose after washing; as such, the observation of residual enzyme activity after washing confirms effective enzyme entrapment. High entrapment efficiencies were also confirmed via washing

experiments in which enzyme activity was assayed in sequential wash solutions; minimal activity losses are observed after the first 10 min wash cycle (which removes poorly entrapped near-surface enzyme), and the printed hydrogel retains >90% of its original activity following 5 h of washing (Supplementary Fig. 6). Furthermore, full substrate conversion occured within 15 min for each entrapped enzyme, demonstrating that the thin printed hydrogel possesses a combination of sufficiently high porosity and low diffusional path length to allow for efficient diffusion of substrate molecules to the enzyme active sites and rapid read-out of enzyme activity.

**Stabilizing enzymes in the printed hydrogel.** In addition to immobilization, the printed thin layer hydrogel has benefits in terms of protecting encapsulated enzymes from proteolytic degradation (via size exclusion), chaotropic agent interference (via competitive hydrogen bonding) as well as minimizing activity losses upon dry storage (via moisture retention). The printed hydrogel prevented proteolytic deactivation of all tested enzymes by proteinase K, with each enzyme retaining >80% of its pre-treatment activity (Figs. 3a–c); in contrast, urease and β-Lac printed directly on the nitrocellulose substrate without the hydrogel retained <10% of their activity over the same treatment time. While the steric barrier presented by the hydrogel is likely the main reason for this result, the ability of the PO-based

hydrogel to resist nonspecific protein adsorption may also be beneficial to reduce the probability of proteinase K binding close to the enzyme. In comparison, co-printing enzymes with POH alone or in combination with unfunctionalized PO also showed only limited benefits in terms of stabilizing both urease and β-Lac against deactivation. AP was a slight outlier in this regard, retaining ~40% of its activity when printed alone and >80% of its activity when printed with POH even in the absence of gel formation (Fig. 3a); this is consistent with the noted high stability of AP relative to other enzymes[46]. However, even with AP, each hydrogel-printed enzyme exhibited better activity retention than any other printed sample tested. Printed hydrogels showed similar efficacy in resisting chaotropic agent-induced denaturation, with hydrogel-printed β-Lac retaining >95% activity following urea challenge compared with only <20% activity retention in solution (Supplementary Fig. 7).

The printed hydrogels are also capable of stabilizing enzymes for long-term storage under (dry) ambient conditions. The hydrogel-entrapped enzymes retained ~100% activity after at least 3 months of storage for AP, urease, and β-Lac (Figs. 3d–f); in contrast, direct printed enzymes lost >70% of their activity within 1 week for urease and β-Lac and within 1 month for AP. While similar efficacy in enzyme stabilization has previously been reported with dried carbohydrate films[47], such films dissolve when placed back in an aqueous environment, leading to rapid

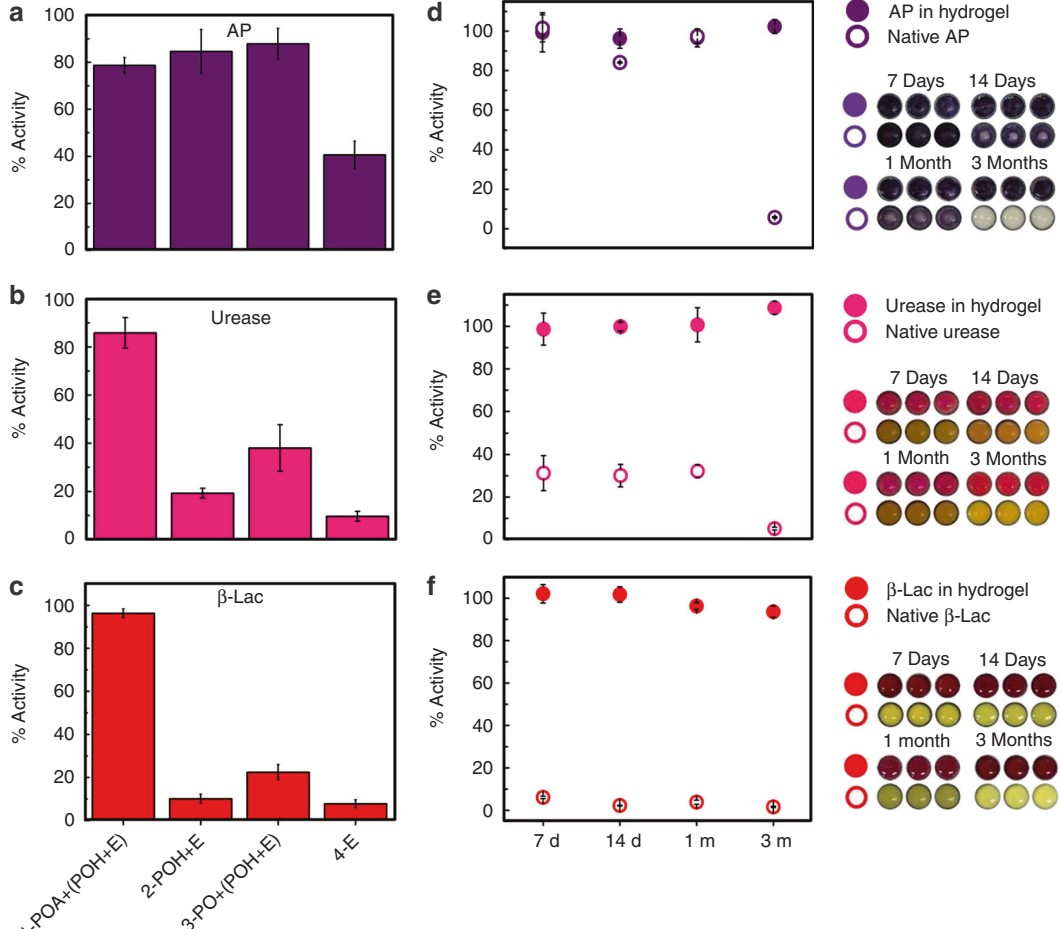

**Fig. 3** Printed hydrogels can stabilize enzymes. Remaining activity of enzymes (E) was quantified (**a**–**c**) after 2 h of protease treatment with proteinase K (a protease that hydrolyzes peptide bonds and thus deactivates enzymes) or (**d**–**f**) following dry storage at room temperature for varying periods of time. The residual activity after each treatment was normalized relative to the initial activity of a freshly printed sample. The hydrogel protects the enzymes against proteolytic degradation by proteinase K and supports enzyme stabilization for long-term storage. (**a**, **d**) Alkaline phosphatase; (**b**, **e**) urease; (**c**, **f**) β-lactamase. Data are presented as means ± SDs (n = 3)

leaching of the enzyme from the substrate. In contrast, the printed hydrogel maintains a confined environment for the enzyme under aqueous conditions, maintaining immobilization while also driving local hydration to promote en zyme activity. Taken together, these results demonstrate that the printed hydrogel has the capacity to protect entrapped enzymes against both degradative agents and time-dependent denaturation, facilitating effective storage and distribution of bioassays developed based on this platform.

**Fabricating a hydrogel-based β-Lac microarray**. To demonstrate the potential of printable hydrogel-enzyme thin films for practical biosensing, TEM-1 β-lac was printed in a POA/POH hydrogel array onto the microzones of a 96-well nitrocellulose wax-printed template, creating a microplate mimic adaptable to current high-throughput screening protocols. Inhibitor solutions and nitrocefin (a colorimetric β-lac substrate) were subsequently deposited on the microzones at different concentrations using a high-throughput dispensing robot. The resulting colorimetric read-out of β-lac activity was quantified via image analysis.

First, well-established true inhibitors of β-lac (tazobactam, sulbactam, and clavulanic acid) were screened to compare $IC_{50}$ values measured via the printed hydrogel assay with solution values determined from a conventional microplate assay. Using the printed hydrogel assay, $IC_{50}$ values of 0.07, 4.1, and 0.15 μM were calculated for tazobactam, sulbactam, and clavulanic acid, respectively (Figs. 4a–c, Table 1); these values compare favorably to the measured solution-based assay $IC_{50}$ values (Figs. 4a–c, Table 1) as well as literature $IC_{50}$ values (Table 1) for these same inhibitors. The excellent quantitative correlation between these results suggests that the printed hydrogel-based assay can determine dose–response relationships of β-lac inhibitors with high accuracy. Furthermore, $IC_{50}$ measurements using the printed hydrogel-based assay require only 5% of the total sample volume used for the solution assay, a significant benefit in screening high-value potential inhibitors (i.e., only 1/20 of the drug candidate mass is required to screen the same drug concentrations). Note also that the potential of the printed hydrogel to stabilize each enzyme enabled 100% activity inhibition at high inhibitor concentrations in each case; in contrast, the increased potential for partial enzyme denaturation in solution over the timescale of the assay led to lower maximum inhibitions in each solution-based assay performed. Lower enzyme concentrations could also be printed to improve the sensitivity of the assay for low $K_i$ inhibitors, with detectable color differences noted between inhibited and non-inhibited printed enzymes at as low as 5 nM enzyme concentrations (Supplementary Fig. 8).

Next, to assess the capacity of the printed hydrogels to differentiate between specific and nonspecific inhibition, the confirmed promiscuous (aggregating) inhibitors rottlerin[19] and BIS IX[19] (both kinase inhibitors) and tetraiodophenolphthalein (TIPT)[17] were tested against TEM-1 β-lac both in solution (modeling a conventional microplate assay) and using a printed hydrogel array. In each case, the aggregating compounds inhibited β-lac in the solution-based assay (a false-positive hit) but were correctly observed to induce no specific inhibition of β-lac in the hydrogel-based assay (Figs. 4d–f). Comparing the aggregate diameter range of ~154–365 nm (Supplementary Fig. 9, Supplementary Table 3) to the ~2 nm characteristic correlation length (i.e., average pore size) of PO hydrogels of this type[48], this performance benefit is likely linked to aggregates not being able to diffuse into the hydrogel and thus being unable to sterically inhibit the enzyme as they can in solution. In this way, the size selectivity of the printed hydrogel layer excludes the promiscuous

inhibitors from accessing the entrapped enzyme and thus avoids the false-positive hits observed in solution assays.

**Discussion**

The printed hydrogel-screening assay reported herein addresses the key limitations associated with current microplate assay platforms for drug discovery by significantly reducing reagent volumes, eliminating costs associated with microtiter plates, and improving assay sensitivity. Sensitivity improvements are gained both in terms of avoiding false hits associated with the aggregation of inhibitor candidates as well as the removal of errors associated with the nonspecific adsorption of enzymes on the surface of polystyrene or polypropylene-based microplates (reported to introduce significant measurement errors for enzymes present in the nanomolar concentration range[49]), the latter leveraging the high protein repellency of PO-based hydrogels[33]. It should be emphasized that these sensitivity improvements were achieved without causing significant increases in screening time, with the full assay (from start to finish) completed within 25 min using only 5% of the total sample volume required for a solution-based assay. The demonstrated ability of the hydrogel platform to discriminate true inhibitors from promiscuous aggregating inhibitors is highly significant, given that promiscuous inhibitors are arguably the most widespread artefact encountered in high-throughput screening. Up to 95% of the active compound hits identified in a typical library consisting of ~70,000 small molecules have been attributed to aggregate-based inhibition[50]. Colloidal aggregates also have a broadly detrimental impact in other pharmaceutical screening contexts, and previously explored techniques to avoid or compensate for such aggregation may denature the enzyme (e.g. detergent-based assays)[21], can alternatively bind to and thus sequester any de-aggregated small molecules (e.g., BSA pretreatments)[23], or cannot be readily transferred to a high-throughput context (e.g., NMR-based assays of self-aggregation)[51]. In contrast, the hydrogel-based screening assay demonstrates minimal interference, the ability to be adapted directly to current high-throughput pharmaceutical screening practices, and exploits a fundamental property of all aggregate-based inhibitors (i.e., size) to exclude their effects on the assay results, strongly suggesting the assay's potential to consistently eliminate false-positive hits. The printing method described herein also offers distinct advantages in the context of enzyme immobilization for a wide range of applications. Covalent conjugation often reduces enzyme activity and does not result in the formation of any kind of physical barrier to block access of colloidal aggregates to the active site; physical adsorption or entrapment of enzymes often results in enzyme desorption and/or denaturation (via charge or hydrophobic interactions) over time. The mixing-induced gelation chemistry used also makes printing much simpler and more reproducible than equilibrium gelation systems like sol-gels, which could otherwise in theory be used in a similar context. As such, the key attributes of this approach (i.e., protein immobilization, protein stabilization, low-volume operation, and elimination of false-positives via steric blocking) are anticipated to be highly beneficial in diverse applications including (but not limited to) medical diagnostics, monitoring the presence of environmental contaminants, or identifying biological warfare agents. In summary, we have demonstrated a printable hydrogel platform that allows for the immobilization, protection, and long-term stabilization of a diverse set of enzymes. Furthermore, using β-lac inhibitors as a model, a low-volume hydrogel-based microarray assay based on this printed gel platform was shown to both accurately measure $IC_{50}$ values of well-established β-lac inhibitors and prevent the occurrence of false-positive hits by promiscuous, nonspecific

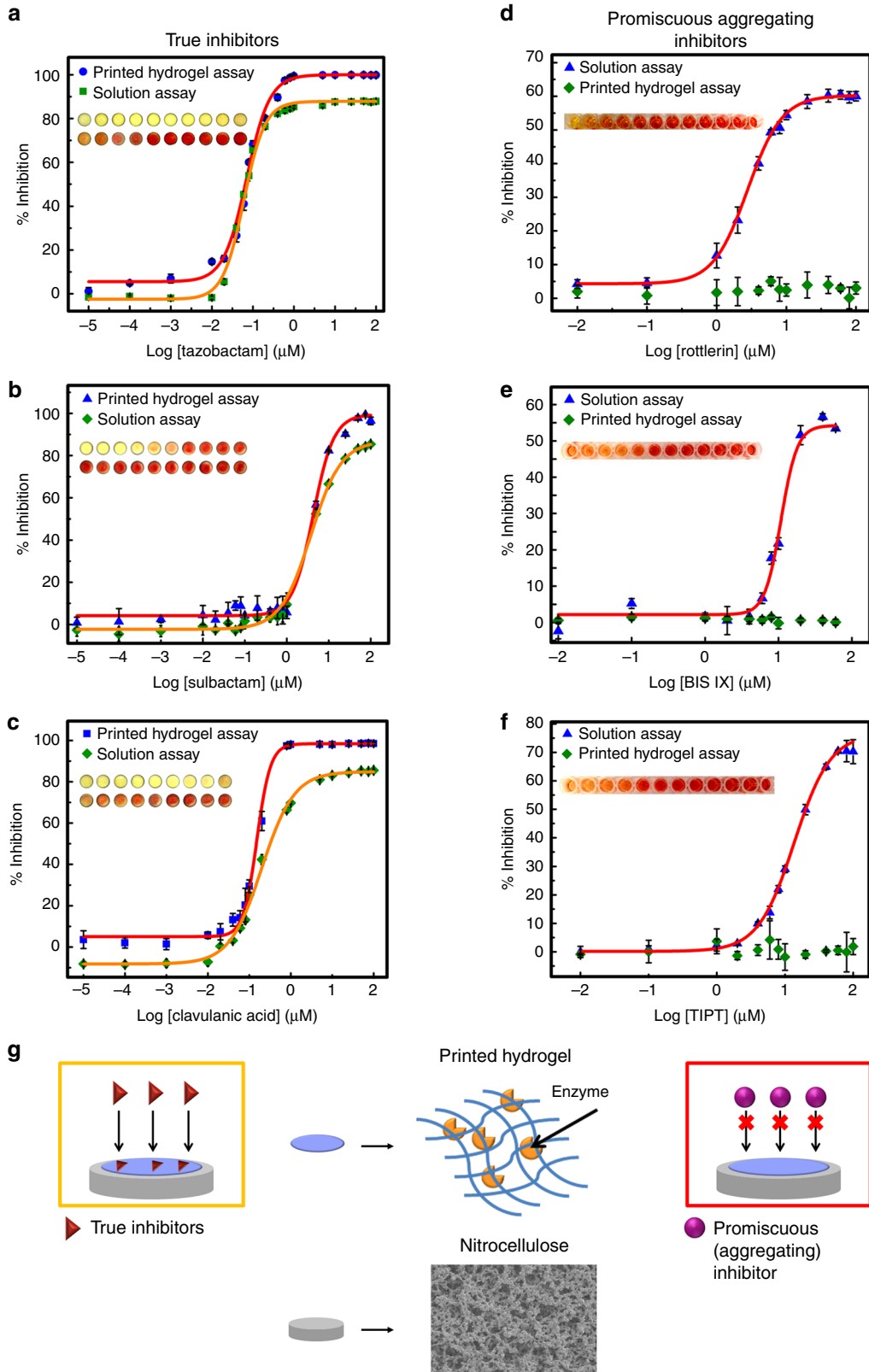

**Fig. 4** A printed hydrogel-based β-lactamase drug-screening assay. **a–c** The printed hydrogel-based β-lactamase screening assay can determine dose–response relationships of classic β-lactamase inhibitors. Comparison of solution vs. printed hydrogel-based inhibition curves for true β-lactamase inhibitors: **a** tazobactam; **b** sulbactam; **c** clavulanic acid. **d–f** The printed hydrogel-based β-lactamase-screening assay can discriminate between true and promiscuous aggregating inhibitors. Comparison of solution vs. printed hydrogel-based inhibition curves for known promiscuous inhibitors of β-lactamase: **d** rottlerin; **e** BIS IX; **f** TIPT. **g** Schematic illustrating how the size selectivity of the printed hydrogel layer excludes the colloid-forming drug (promiscuous aggregating inhibitor) from accessing the encapsulated enzyme but can permit the diffusion of a soluble drug (true inhibitor) to generate a positive signal for true inhibitors only. Data are presented as means ± SDs ($n = 3$)

**Table 1 Comparison of IC$_{50}$ values of classic β-lactamase inhibitors IC$_{50}$ values of tazobactam, sulbactam, and clavulanic acid measured by the printed hydrogel assay relative to the conventional solution assay and reported literature values**

| β-Lactamase inhibitor | IC$_{50}$ (µM) | | |
|---|---|---|---|
| | Printed hydrogel assay | Solution assay | Literature[52] |
| Tazobactam | 0.07 ± 0.01 | 0.06 ± 0.01 | 0.04 |
| Sulbactam | 4.1 ± 0.2 | 4.0 ± 0.3 | 6.1 |
| Clavulanic acid | 0.15 ± 0.01 | 0.19 ± 0.01 | 0.09 |

Values are represented as mean ± SD (n = 3)

inhibitors that function via aggregation instead of a specific binding interaction. Combining the functionality of the assay with the facile printing of multisample substrates amenable to use in current high-throughput screening formats, this approach offers potential to significantly improve the resolution of drug lead optimization testing and thus streamline the drug discovery process. The ease of fabrication, potential for long-term stabilization of labile biomolecules both in solution and in dry storage, and the versatile functionality of this platform make this approach promising not only for drug discovery but also in a range of other sensing and analytical applications.

## Methods

**Hydrogel printing**. A nitrocellulose microzone plate was fabricated by printing hydrophobic wax barriers onto a nitrocellulose membrane (EMD Millipore) using a Xerox ColorQube 8570 N solid wax printer and a 96-well-plate template (3 mm diameter wells, ~9 mm inter-well distance). The wax-printed paper was placed into an oven at 120 °C for 2 min to melt the wax through the paper. Polymer inks comprising 6 w/w% aldehyde-functionalized POA or POH (see Supplementary Methods for details on synthesis and characterization) were printed in 10 mM PBS containing 5 w/w% glycerol as a humectant and viscosity modifier; the resulting viscosities of the POA and POH inks were 3.27 and 4.85 mPa·s, respectively. A BioJet HR™ non-contact solenoid dispenser was used to print the inks on the paper microzones. The two reagent lines were charged with POA and POH, the dispenser valve was programmed to stay open for 6 ms, and the frequency was set to 100 Hz. The thin-layer hydrogel was fabricated by dispensing 2 µl POA onto the microzone, immediately followed by 2 µl POH. The samples were dried and stored at room temperature.

**Enzyme entrapment studies**. POH ink solutions containing one of the tested model enzymes were prepared with enzyme concentrations listed in Supplementary Table 1. Entrapped protein samples were printed as previously described, followed by washing with 10 mM PBS at 300 rpm on an IKA MS3 Basic Shaker for 10 min. The relevant substrate solutions for each enzyme were then pipetted onto the washed samples using the solutions and volumes listed in Supplementary Table 2 to assess enzyme activity. Images of the resulting colorimetric read-out were taken with an IPhone 5C camera. Image analysis to determine colorimetric intensity was performed using Fiji, an open-source program based on ImageJ. The converted substrate color was extracted using the Color Deconvolution plugin. Extracted images were inverted and converted to 8 bit grayscale images. The intensity of each sample was measured and presented as a ratio of the corresponding control image (a sample printed with the same concentration of enzyme but not washed to remove any non-adsorbed enzyme).

**Protease protection studies**. An aliquot of 10 µl of a 2 mg ml⁻¹ proteinase K solution (prepared in 10 mM PBS and 1 mM CaCl$_2$) was pipetted on the printed enzyme samples both with and without hydrogel encapsulation. The samples were incubated in a closed container for 2 h at room temperature, after which substrate solutions were pipetted on the treated samples at the volumes listed in Supplementary Table 2. Image acquisition and analysis was performed as described for the entrapment studies. The intensity of each sample was measured and presented as a ratio of the corresponding control image (the same hydrogel/enzyme combination not treated with protease).

**Long-term stability studies**. Printed enzyme samples were stored in a closed, dark container at room temperature for varying periods of time. Image acquisition and

analysis was performed as described previously for the entrapment and proteinase K degradation studies. The intensity of each sample was measured and presented as a ratio of the corresponding control image (freshly printed using the same hydrogel/enzyme combination).

**Solution-based β-lac assay**. True inhibitor (tazobactam, sulbactam, and clavulanic acid) solutions were prepared in DIW, and promiscuous inhibitor (rottlerin, BIS IX, and TIPT) solutions were diluted in DIW from 10 mM DMSO stock solutions. The assay mixture contained 25 nM β-lac and a range of inhibitor concentrations relevant to the IC$_{50}$ of the true inhibitors and the apparent IC$_{50}$ of the aggregating promiscuous inhibitors in 100 µl of 10 mM PBS buffer. β-lac and inhibitor were pre-incubated for 10 min, after which nitrocefin was added to a final concentration of 200 µM. β-lac activity was then assessed via UV-vis spectrophotometry by tracking the hydrolysis of nitrocefin by monitoring solution absorbance at 492 nm (Infinite M1000 spectrophotometer, Tecan).

**Printed hydrogel-based β-lac assay**. POH ink solution was prepared with a final concentration of 50 nM β-lac and used to print hydrogel spots on a 96-well paper microzone plate as described previously. An aliquot of 5 µl of tazobactam, sulbactam, and clavulanic acid solutions (at starting concentrations of 100 µM) were added to each microzone using a Tecan Freedom Evo 200 liquid-handling robot (Tecan, Switzerland). The inhibitor was incubated with the printed β-lac for 20 min, after which the assay was initiated with the addition of 5 µl of nitrocefin (500 µM) to each microzone. A similar protocol was used to test the promiscuous inhibitors rottlerin, BIS IX, and TIPT, again using starting concentrations of 100 µM. Images were taken with a Canon DSLR camera (operated in manual focus mode without flash) after 25 min. The wax-printed background was filtered out using GIMP software (Version 2.8.16). Image analysis was performed using Fiji, with the converted substrate color extracted using the Color Deconvolution plugin. Extracted images were inverted and converted to 8 bit grayscale images. The intensity of each sample was measured and presented as a ratio of the control image (i.e., a microzone not treated with inhibitor). Calculation of IC$_{50}$ values was carried out in OriginPro by plotting the calculated percentage inhibition against the added inhibitor concentration. Curve fitting was performed with the dose–response function (OriginLab Corporation, Northampton, MA, USA).

**Data availability**. All relevant data are available from the authors upon request.

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

## Acknowledgements

Glynis de Silveira (Canadian Center for Electron Microscopy, McMaster University) is acknowledged for acquiring the SEM images, while Danielle Covelli (Biointerfaces Institute, McMaster University) is thanked for collecting the XPS data. The staff of the Biointerfaces Institute is thanked for their support in completing this project. Dr. Carlos Filipe is thanked for his critical reading of the manuscript. Funding from the Natural Sciences and Engineering Research Council of Canada (NSERC) is gratefully acknowledged.

## Author contributions

R.M., M.M.A. and T.H. designed the project. R.M. performed and analyzed all experiments, except for collecting SEM images and XPS measurements. R.M. and T.H. wrote the manuscript.

## Additional information

**Competing interests:** The authors declare no competing financial interests.

