## [Peer Review File · Nature Communications]

Reviewers' comments:

Reviewer #1 (Remarks to the Author):

The major claims of the paper is the production of a printable hydrogel matrix, without the use of UV light, to immobilize enzymes for the query of specific inhibitors. The work is important as some enzyme inhibitors are non-specific through physical interactions that may not be relevant in fields such as antibiotic discovery.

The work is fairly convincing but not particularly surprising. The authors discuss the pitfalls of certain immobilization techniques such as cross-linking or covalent attachment but offer no comparison of these metrics with the reported technology. The technology is only compared to solution based systems. The authors should either design experiments for comparative purposes or provide ample discussion explaining what evidence demonstrates an improvement over other immobilization techniques.

In figure 2, the authors demonstrate that the enzyme does not have good residual activity by itself or with the POH monomer post-washing but does not provide data for deposition with just the POA monomer and enzyme. This should be examined.

In figure 4, the authors demonstrates excellent data that the enzyme does not lose activity to aggregation agents. However, it is odd that the hydrogel immobilized enzymes never reach 100% inhibition in the presence of the true inhibitors as compared to the solution enzymes which do reach 100% inhibition. Can the authors comment on this effect?

Reviewer #2 (Remarks to the Author):

The manuscript by Mateen et al. presents an interesting application of hydrogels for the measuring of enzyme activity, on that is tested against three widely studied enzymes. The authors argue persuasively that the method can be used to screen for enzyme modulators in high throughput with only 5% of the materials, while the hydrogel by its nature removes false positives arising from colloidal aggregation. Their examples seemed convincing.

I found little wrong with this manuscript and broadly support its publication. The one quibble I might make is that 25 nM seems like a very high concentration for an enzyme like TEM-1, which has been described as a "perfect enzyme", and certainly has very high turn-over rates. Is it the nature of the hydrogel that demands such high enzyme concentrations, and does that play out for other enzymes? It's not so much a question of using too much enzyme, as the amounts are reduced already, but rather limiting the best K_i that can be detected, as getting K_i values below that of the enzyme will be hard, especially in high-throughput, and even as the K_i approaches the $[E]$ strange things will begin to happen to the Hill slope, for instance.

Notwithstanding this quibble, I thought this paper will be of interest to the community, and it may help address several problems in the field.

Reviewer #3 (Remarks to the Author):

In this manuscript, authors prepared hydrogel-based enzyme microarray for drug screening. Hydrogel was formed by sequential printing of POH and POA, reaction of which forms the gels via hydrazone bond formation. By entrapping enzymes within hydrogel, various benefits could be obtained such as better stability against proteolytic degradation, avoiding false false signal and storage. The techniques utilized in the investigations are straightforward and this reviewer does not have any major technically related issues. However, much of the results are anticipated in this study and this reviewer could not find enough novelty and significance of this work. This reviewer believes that it is better to submit this manuscript to more specific journals related to biosensing

1. Prof. Hamachi group published hydrogel-based protein microarray in Nature Materials (2004). I could not find significant advance in this study over previous work. Also this paper should be included in the reference.

2. Using layer-by-layer printing does not look efficient. In this study, enzymes were entrapped in the upper layer. How about entrapping enzyme in bottom layer? Authors should provide the thickness of hydrogel layer and distribution of enzyme within the hydrogel.

3. Did enzyme leach out from the hydrogel? Can authors control the porosity of hydrogel so that their mesh size was large enough to allow the diffusion of target and small enough to prevent enzyme leaching?

4. In discussion, authors explained the general advantages of using hydrogel over direct enzyme immobilization on the surface such as preventing non-specific adsorption and high water content for preserving enzyme activity, which have been already well-known. Comparisons with enzyme entrapment inside hydrogel using UV polymerization will be more helpful in persuading the excellence of this study.

Todd Hoare, Ph.D.

Canada Research Chair in Engineered Smart Materials (Tier 2)
University Scholar and Associate Professor
Department of Chemical Engineering, McMaster University
1280 Main St. W., Hamilton, ON, Canada L8S 4L7
Phone: 1-905-525-9140 ext 24701 | E-mail: hoaretr@mcmaster.ca | Website: <http://hoarelab.mcmaster.ca>

Reviewer(s)' Comments to Author:

Reviewer #1 (Remarks to the Author):

The major claims of the paper is the production of a printable hydrogel matrix, without the use of UV light, to immobilize enzymes for the query of specific inhibitors. The work is important as some enzyme inhibitors are non-specific through physical interactions that may not be relevant in fields such as antibiotic discovery.

The work is fairly convincing but not particularly surprising. The authors discuss the pitfalls of certain immobilization techniques such as cross-linking or covalent attachment but offer no comparison of these metrics with the reported technology. The technology is only compared to solution based systems. The authors should either design experiments for comparative purposes or provide ample discussion explaining what evidence demonstrates an improvement over other immobilization techniques.

Response: The reviewer is correct in stating that this is hardly the only immobilization technique available for enzymes, as this has been an active area of research for years. However, aside from this paper representing to our knowledge the first report of successful printing of *in situ*-gelling covalently crosslinked hydrogels on a substrate (a significant and novel technical advance in its own right), the key point of novelty in this paper is not the immobilization method itself but rather the *combination* of the immobilization strategy *plus* the size selectivity of the hydrogel to ensure effective enzyme entrapment, as well as access of soluble drugs but exclusion of colloidal aggregates to the active site. Simply put, without a hydrogel to provide the porosity necessary for size selectivity, this technology does not work, and other typical immobilization techniques applied cannot replicate these benefits. In this context, performing additional immobilization comparisons in our view is neither necessary nor instructive regarding the essential benefits of the system - metrics such as immobilized enzyme efficiency that can be used to compare immobilization techniques do not tell the whole story necessary to support the application we disclose. That being said, the printing method we disclose herein offers distinctive advantages in the context of enzyme immobilization for the particular application investigated. Covalent attachment of enzymes to adsorbent resins is often used for the synthesis of industrially relevant reagents¹, with such attachment noted to typically increase the half-life and thermal stability of the enzyme.² However, this requires a chemical conjugation step that can reduce activity and, in the context of the application we demonstrate, would not pose a physical barrier to block access of colloidal aggregates to the active site. Physical entrapment of enzymes can also be used, but often results in dissolution of the enzyme back into solution when placed back in buffer, can be difficult to process and print (e.g. sol-gel systems, which could at least in theory be used in a similar context), and can also lead to problems with protein denaturation (e.g. charge complexation).³ The hydrogel used in this work shows substantial benefits in maintaining high water content around the enzyme without denaturation, even in dry conditions over extended storage times. In addition, this is the first time that an enzyme immobilization technique of any kind has to our knowledge been used to address the challenge of discriminating between true and promiscuous inhibitors in high throughput drug screening, a highly novel and significant contribution that offers strong potential to accelerate the drug discovery process. While the result may not be “surprising”, we would strongly argue that the combination of the new printing process and the unique idea to use hydrogel size selectivity to solve a significant and costly problem in the pharmaceutical industry strongly supports the novelty and significance of the paper.

In figure 2, the authors demonstrate that the enzyme does not have good residual activity by itself or with the POH monomer post-washing but does not provide data for deposition with just the POA monomer and enzyme. This should be examined.

Response: We originally did not do this experiment since it is our general approach to minimize contact between the aldehyde-functionalized polymers (prior to gelation) and proteins, which may form Schiff base interactions with the aldehyde groups and result in some degree of protein denaturation. When printed in a gel in which the majority of aldehyde groups are consumed via gelation, we have not seen this problem in any of our previous work (or in this work). However, we agree with the reviewer that the omission of this data was an oversight in the original manuscript to justify the process chosen. In response, we have performed this experiment and added the relevant data to Fig. 2c-e where the POH and PO co-printing results were already shown. The data, shown in Figure R1 below, indicates significant loss of enzyme activity (>70%) upon printing all three enzymes with POA only, as anticipated. We have also edited the text to indicate the inclusion of the POA data (p. 10):

“In comparison, co-printing enzymes with POH alone, POA alone, or in combination with unfunctionalized PO showed only limited benefits in terms of immobilizing and stabilizing the enzymes. (Figs. 2c-e).”

Figure R1 | Post-washing activity of printed enzymes in POA ink. Residual activity of alkaline phosphatase (AP), urease and β -lactamase (β -Lac) printed in POA ink, following washing of samples in 10 mM PBS for 10 min relative to the corresponding unwashed control. The three adjacent images in each series are replicates. Error bars represent the standard deviation from the mean ($n=3$).

In figure 4, the authors demonstrates excellent data that the enzyme does not lose activity to aggregation agents. However, it is odd that the hydrogel immobilized enzymes never reach 100% inhibition in the presence of the true inhibitors as compared to the solution enzymes which do reach 100% inhibition. Can the authors comment on this effect?

Response: We think the reviewer may have misread the figure, which actually shows the opposite. The inhibition of enzymes in the *solution-based* assay does not reach 100% when treated with a true inhibitor (Fig. 4, see green data points). This is sometimes observed in solution-based assays because of the potential partial denaturation of the enzyme in its unprotected state in solution during the course of the assay, particularly in the presence of a drug that may hydrophobically or electrostatically interact with the protein; in such cases, the altered conformation of the enzyme can result in even true inhibitors not fully being able to block the activity of the enzyme. However, the inhibition of *immobilized* enzymes in the *printed hydrogel* assay does reach 100% when treated with a true inhibitor (Fig. 4, see blue data points), consistent with the hydrogel providing better stabilization for the enzyme than is provided in solution. Indeed, this represents another benefit of the use of the printed hydrogel in this application, as the gel helps to maintain full enzyme activity during the testing protocol. We have added text to support this argument in the manuscript (p. 15):

“Note also that the potential of the printed hydrogel to stabilize each enzyme enabled 100% activity inhibition at high inhibitor concentrations in each case; in contrast, the increased potential for partial enzyme denaturation in solution over the timescale of the assay led to lower maximum inhibitions in each solution-based assay performed.”

Reviewer #2 (Remarks to the Author):

The manuscript by Mateen et al. presents an interesting application of hydrogels for the measuring of enzyme activity, on that is tested against three widely studied enzymes. The authors argue persuasively that the method can be used to screen for enzyme modulators in high throughput with only 5% of the materials, while the hydrogel by its nature removes false positives arising from colloidal aggregation. Their examples seemed convincing.

I found little wrong with this manuscript and broadly support its publication. The one quibble I might make is that 25 nM seems like a very high concentration for an enzyme like TEM-1, which has been described as a "perfect enzyme", and certainly has very high turn-over rates. Is it the nature of the hydrogel that demands such high enzyme concentrations, and does that play out for other enzymes? It's not so much a question of using too much enzyme, as the amounts are reduced already, but rather limiting the best K_i that can be detected, as getting K_i values below that of the enzyme will be hard, especially in high-throughput, and even as the K_i approaches the $[E]$ strange things will begin to happen to the Hill slope, for instance.

Notwithstanding this quibble, I thought this paper will be of interest to the community, and it may help address several problems in the field.

Response: The reviewer is correct in stating that the specific enzyme concentration (particularly for TEM-1 β -Lac) has high turn-over rates. For the purposes of the assay shown, we selected the concentration of TEM-1 β -Lac based on the level of colour contrast between microzones treated with and without a true inhibitor. While this selection was done by trial-and-error for the original manuscript, we have now systematically printed different enzyme concentrations within the hydrogel and subsequently treated each spot with 100 μ M tazobactam or water for 20 min, followed by the addition of the substrate to each microzone over an additional 25 min. In the now-added Supplementary Figure S8 (reproduced below for convenience), while the greatest amount of colour contrast was observed when 25 nM of enzyme was printed on each microzone, detectable colour differences between the non-inhibited (left) and inhibited (enzyme) activities were also noted down to 5 nM enzyme concentrations. As such, in terms of fabricating a commercial version of this prototype, there is flexibility to reduce the enzyme concentration (particularly for high turn-over enzymes) and address the issue posed by the reviewer. We have now noted this added flexibility of the assay in the text of the revised manuscript (p. 16):

“Lower enzyme concentrations could also be printed to improve the sensitivity of the assay for low K_i inhibitors, with detectable colour differences noted between inhibited and non-inhibited printed enzymes at as low as 5 nM enzyme concentrations (Supplementary Figure S8).”

Figure S8 | Optimizing the β -lactamase concentration in a printed hydrogel-based screening assay. A range of β -lactamase concentrations was encapsulated via printing in the hydrogel, with the colorimetric readouts compared with and without tazobactam (100 μ M) treatment. Significant and detectable colour differences were observed using enzyme concentrations as low as 5 nM, providing additional flexibility for the assay in detecting low K_i inhibitors.

Reviewer #3 (Remarks to the Author):

In this manuscript, authors prepared hydrogel-based enzyme microarray for drug screening. Hydrogel was formed by sequential printing of POH and POA, reaction of which forms the gels via hydrazone bond formation. By entrapping enzymes within hydrogel, various benefits could be obtained such as better stability against proteolytic degradation, avoiding false false signal and storage. The techniques utilized in the investigations are straightforward and this reviewer does not have any major technically related issues. However, much of the results are anticipated in this study and this reviewer could not find enough novelty and significance of this work. This reviewer believes that it is better to submit this manuscript to more specific journals related to biosensing

Response: As noted in our response to Reviewer #1, we respectfully disagree that this paper has insufficient novelty or significance for publication in *Nature Communications*. In terms of novelty: (1) this is the first example of printing of covalent *in situ*-gelling hydrogels, opening the door for several new types of printing applications for protein-passivating, encapsulating, or active coatings; (2) this is the first example of a high-throughput assay capable of discriminating specific versus nonspecific inhibitors,

directly addressing a critical bottleneck in drug discovery; (3) the combination of the printability of the assay with the capacity of the hydrogel coating to both stabilize/immobilize the enzyme and provide size-selective screening of colloidal vs. soluble inhibitors makes this approach uniquely adaptable to existing high-throughput screening instrumentation and techniques while also significantly lowering the required sample volumes. Collectively, we would argue these contributions represent the synthesis of several ideas (both new and established) to form a functional device which is highly significant. As such, while the key enabling physical realities that hydrogel porosity can be tuned to exclude certain sizes and enzymes can be immobilized on a substrate are certainly not new or surprising, the way we have implemented them to form a new technology is in our view highly novel.

In terms of the significance of the problem, high-throughput screening is inarguably an indispensable tool for the identification of new drug candidates. However, the relatively high volume of drug candidate required for analysis and the poor resolution of detection methods with current high throughput techniques clearly warrant a critical look at the current high-throughput screening platform. In particular, the problem of accurately discriminating aggregating (or promiscuous) inhibitors (which represent up to 95% of the “hits” in a conventional drug screen) represents a significant time and economic bottleneck in the drug discovery process that requires a solution in order to accelerate new drug discovery. We think that this paper convincingly shows one such solution which is implementable within a conventional drug screening geometry and workflow, making the contribution of high significance.

1. Prof. Hamachi group published hydrogel-based protein microarray in Nature Materials (2004). I could not find significant advance in this study over previous work. Also this paper should be included in the reference.

Response: We have, at the reviewer’s request, included this paper as a reference, as it is a very elegant example of a protein microarray for drug screening. However, we respectfully disagree with the reviewer that our work has not advanced on this paper. There are several key differences in terms of the gel itself and how the array was fabricated. In Hamachi’s paper, a supramolecular hydrogel was formed from a low-molecular weight hydrogelator, a glycosylated amino acid scaffold that gels in water at low concentrations via a combination of hydrogen bonding, van der Waals, and hydrophobic interactions; all of these interactions are significantly less stable than the covalent gelation strategy implemented in this work. Moreover, the hydrogel was amphiphilic as it contained both hydrophobic domains and an aqueous cavity, a structure that can pose problems with protein stability over long time frames; in contrast, our hydrogel does not contain any hydrophobic domains whatsoever. A one-hour aging step was required to complete the fabrication; our hydrogel is printable at concentrations that promote rapid gelation within seconds upon sequential deposition at the substrate, greatly improving the potential for high-throughput scale-up. Our printed volumes are also much lower, enabling the use of less enzyme and less drug for screening. Hamachi’s protein loading technique was also substantially different, with 1 μ L of a protein solution injected into each gel spot and incubated at room temperature for 15 min. (i.e. loading is via diffusion post-gelation); in contrast, we directly mix the enzyme in the hydrogel ink to create a uniform distribution of encapsulated enzyme inside the printed gel within seconds all in a single printing step.

The actual assay developed by Hamachi also has clear differences with what we report. While Hamachi’s assay could report whether or not an enzyme was active or inhibited, it was not used to calculate IC_{50} values (as essential for drug screening). **Most critically, it was not demonstrated to offer**

selectivity between real versus promiscuous inhibitors to prevent false hits. This latter point is the key point of functional novelty of our device, which has never before been reported in this or any other paper in the literature to our knowledge. In addition, the hydrogel microarray developed is not directly adaptable to existing high-throughput screening practices. As such, while Hamachi's paper is clearly one of the pioneering papers within this field (and deserves a citation in our paper), there are very clear structural and functional differences between what is reported and what we report in this manuscript. In this context, we remain confident of the novelty and significance of the results we present.

2. Using layer-by-layer printing does not look efficient. In this study, enzymes were entrapped in the upper layer. How about entrapping enzyme in bottom layer? Authors should provide the thickness of hydrogel layer and distribution of enzyme within the hydrogel.

Response: First, we disagree with the reviewer that the printing process is inefficient. Particularly relative to Hamachi's work described above (in which gelation requires multiple steps timed over the course of one hour, plus an additional 15 minute step for protein loading), we can print a full 96-well microarray sheet by directly encapsulating protein within the gel in minutes with our current system. There is no obvious limitation to scale this up to more efficient and high-throughput printing processes if desired, provided the chosen printing process enables a single spot on the substrate to be printed twice.

Second, the enzyme is not actually entrapped only in an upper layer, as the reviewer suggests. Indeed, gelation is impossible unless the first and second layers fully mix – otherwise, the free precursor solution would simply wash away once placed back in water. While the initially printed POA ink is dry when the POH + enzyme active ink is printed, the water in the POH ink re-hydrates the highly soluble POA ink to provide physical mixing of the components, gelation, and a uniform distribution of the enzyme throughout the printed hydrogel layer. To demonstrate this explicitly, we have performed confocal microscopy on printed hydrogel microzones containing 0.005 mg/mL FITC-BSA and rhodamine-labeled POA (the base polymer layer, into which the reviewer is implying the protein could not mix), the results of which are shown in the new Supplementary Figure S5. FITC-BSA was distributed remarkably evenly throughout the 80 μM thickness (and 326 x 326 μm area) of the imaged area, confirming that printed BSA is not localized only at the surface of the hydrogel film. Similarly, POA-rhodamine is found at high concentrations at the top of the printed hydrogel even though it was printed first, confirming the interdiffusion of the printed polymers upon printing of a secondary polymer layer. We have added the following text to the manuscript to emphasize this finding (p. 10):

“Fluorescence microscopy images of printed FITC-BSA confirmed the uniform distribution of the protein on the nitrocellulose surface when entrapped in the thin layer hydrogel (Fig. 2b), **while confocal microscopy images of FITC-BSA encapsulated inside a hydrogel prepared with Rhodamine-POA confirm that the printed protein is distributed evenly throughout both the cross-section and the depth of the printed hydrogel microzones (Supplementary Fig. S5).**

Figure S5 | Cross-sectional confocal microscopy images of printed hydrogel microzones demonstrate that FITC-BSA is distributed throughout the thickness of the hydrogel film. FITC-BSA channel, Rhodamine-POA channel and overlaid fluorescence images confirm the co-localization of FITC-BSA within the gel as well as the relatively uniform distribution of FITC-BSA within the printed gel. The printed hydrogel was prepared using 0.005 mg/mL FITC-BSA in the POH ink and rhodamine-labeled POA as the base polymer. Top (a) and bottom (b) views of the 326 x 326 μm cross-sectional slice imaged at a depth of 80 μm are presented.

3. Did enzyme leach out from the hydrogel? Can authors control the porosity of hydrogel so that their mesh size was large enough to allow the diffusion of target and small enough to prevent enzyme leaching?

Response: We can absolutely adjust the porosity of the hydrogel to enable enzyme entrapment but high diffusibility of our target; indeed, as previously described in this response, this is one of the key attributes of our technology that makes it work for discriminating true versus promiscuous inhibitors. The diffusibility of our target is clearly evident in the inhibition results shown as well as the preservation of fast assay times (<15 minutes for full substrate conversion); neither of these observations are possible without relatively free diffusion of the true inhibitors into the hydrogel. In terms of enzyme entrapment, we showed in the original manuscript that alkaline phosphatase (AP), urease (Ur), and β -lactamase (β -Lac) were effectively immobilized and stabilized in the printed hydrogel with >90% activity maintained for AP and β -Lac and >85% activity maintained for urease relative to enzymes printed in the same manner but not rinsed prior to activity testing (See Fig. 2c-e in manuscript). To further demonstrate that the printed hydrogel is effective for enzyme encapsulation, β -lactamase activity was monitored in solutions that were used to wash samples of hydrogel printed β -lactamase, with the enzyme activity compared to the observed reading in a buffer-only control sample. This data has been added to the Supplementary Information as **Fig. S6**. Minimal (leached) enzyme activity was observed in the 10 min. wash solution, likely due to a small fraction of enzyme localized near the surface of the hydrogel film being poorly entrapped within the printed hydrogel and thus prone to minor leaching. Almost no enzyme activity was observed in any of the wash solutions collected at later time points. Collectively, these leachates represent <10% of the total enzyme printed, a negligible amount of leaching that is inevitable to at least some extent with any printed gel system (and significantly less than would be observed with any post-printing loading strategy). Furthermore, hydrogel printed β -lactamase samples washed for five hours retained >90% of the activity of a fresh hydrogel-entrapped sample (inset of **Fig. S6**), confirming that the total amount of enzyme lost in the washing steps via leaching is very small. As such, we can achieve appropriate pore sizes in our hydrogel to facilitate high enzyme entrapment but also high access of the inhibitor candidates to the enzyme active site. We have added the following text to the manuscript to emphasize this finding (p. 10):

“High entrapment efficiencies were also confirmed via washing experiments in which enzyme activity was assayed in sequential wash solutions; minimal activity loss is observed after the first 10 minute wash cycle (which removes poorly entrapped near-surface enzyme), and the printed hydrogel retains >90% of its original activity following five hours of washing (Supplementary Fig. S6).”

Figure S6 | Printing β-lactamase in a hydrogel minimizes enzyme leaching. Printed samples were washed in 10 mM PBS for varying amounts of time. β-lactamase activity was then assessed in the wash solutions via UV-vis spectrophotometry by tracking the hydrolysis of nitrocefin by monitoring solution absorbance at 492 nm. The resulting absorbance readings are reported as a ratio of the control (i.e. the absorbance of buffer itself at 492 nm). Residual activity of samples washed for 5 h relative to the corresponding unwashed control is presented in the inset graph, confirming that minimal quantities of enzyme are leached from the printed hydrogel. Error bars represent the standard deviation from the mean ($n=3$).

4. In discussion, authors explained the general advantages of using hydrogel over direct enzyme immobilization on the surface such as preventing non-specific adsorption and high water content for preserving enzyme activity, which have been already well-known. Comparisons with enzyme entrapment inside hydrogel using UV polymerization will be more helpful in persuading the excellence of this study.

Response: Again, we absolutely agree that this is not the first example of using a hydrogel to immobilize an enzyme on a surface to reduce non-specific adsorption and maintain high water content – we are not at all suggesting that this is the key novel feature of our paper (see our response to Reviewer 3’s general comments at the start of the response). The printing system we have developed is particularly advantageous since it is amenable to dispensing small volumes (minimizing sample volumes for screening), can localize materials in specific patterns on a variety of different substrates (enabling, for example, facile printing of multi-sample arrays on a substrate), and can be scaled to commercial production. UV photopolymerization represents a post-processing step that would make scaling the printing significantly more challenging. Furthermore, UV photopolymerization can have negative effects on biological gel cargo as the energy of the polymerizing light, the potential toxicity of the photoinitiators, monomers, and/or photoinitiator by-products, as well as the radical species generated

during polymerization may damage the entrapped biomolecules. We have clarified our argument on this point in the revised paper (p. 7), including a new reference inserted to support our argument:

“To our knowledge, this method represents the first example of a printed covalently crosslinked hydrogel that does not require UV photopolymerization to facilitate gelation⁴², **a process in which the energy of the polymerizing light, the potential cytotoxicity of the photoinitiators/monomers (both during processing as well as residual within the final gel), as well as the radical species generated during polymerization have in some cases been reported to damage the entrapped biomolecules⁴**

Response to reviewers references

1. Damnjanović, J.J. et al. Covalently immobilized lipase catalyzing high-yielding optimized geranyl butyrate synthesis in a batch and fluidized bed reactor. *Journal of Molecular Catalysis B: Enzymatic* **75**, 50-59 (2012).
2. Mohamad, N.R., Marzuki, N.H.C., Buang, N.A., Huyop, F. & Wahab, R.A. An overview of technologies for immobilization of enzymes and surface analysis techniques for immobilized enzymes. *Biotechnology & Biotechnological Equipment* **29**, 205-220 (2015).
3. Liang, J.F., Li, Y.T. & Yang, V.C. Biomedical application of immobilized enzymes. *Journal of pharmaceutical sciences* **89**, 979-990 (2000).
4. Baroli, B. Photopolymerization of biomaterials: issues and potentialities in drug delivery, tissue engineering, and cell encapsulation applications. *Journal of Chemical Technology and Biotechnology* **81**, 491-499 (2006).

REVIEWERS' COMMENTS:

Reviewer #1 (Remarks to the Author):

The responses to the reviewer comments were well thought out and explained. I do think that the direct comparison between other immobilization techniques and the presented one that was included in the response should be added to the discussion section for additional clarity. However, I am still wary of the novelty despite the explanation. Use of size exclusion hydrogels is a well known concept, and its application in this context does seem better suited for more specialized journals focused on biosensing or high-throughput screening.

Reviewer #2 (Remarks to the Author):

The authors have carefully addressed my critique about high enzyme concentration, and I am satisfied with their response. I would note that it is now clear that they do lose signal to noise in this format--as for plate-based aqueous screens, for instance, far less TEM-1 can be used--and I think that this caveat could be strengthened even slightly further, as it may affect all enzymes. Still, this is a quibble and I certainly support publication of this manuscript. Their responses to the other reviewers also seemed thoughtful and represent substantial work, though naturally I will defer to the other reviewers here.

Still, certainly from my standpoint, the manuscript can be published. I think it will find an interested readership.

Reviewer #3 (Remarks to the Author):

The authors have made great efforts to persuade reviewers. I agree with many of the parts described by the author. However, as reviewer 1 point out, quantitative comparison with other hydrogel systems than solution assays will make this manuscript more strong

Although UV polymerization is cytotoxic and poor, many studies have already encapsulated and cultured cells (even stem cells) with photocrosslinking. The UV polymerization system mentioned by the authors seems to be an example of the worst case. It would be wrong to mention that photocrosslinking is unconditionally bad.

Response to Reviewers:

Reviewer #1 (Remarks to the Author): The responses to the reviewer comments were well thought out and explained. I do think that the direct comparison between other immobilization techniques and the presented one that was included in the response should be added to the discussion section for additional clarity. However, I am still wary of the novelty despite the explanation. Use of size exclusion hydrogels is a well known concept, and its application in this context does seem better suited for more specialized journals focused on biosensing or high-throughput screening.

Response: We have added the following text (adapted from our response document from the last cycle of reviews) to the end of the discussion section of the main manuscript as per the reviewer's suggestion. The printing method described herein also offers distinct advantages in the context of enzyme immobilization for a wide range of applications. Covalent conjugation often reduces enzyme activity and does not result in the formation of any kind of physical barrier to block access of colloidal aggregates to the active site; physical adsorption or entrapment of enzymes often results in enzyme desorption and/or denaturation (via charge or hydrophobic interactions) over time. The mixing induced gelation chemistry used also makes printing much simpler and more reproducible than equilibrium gelation systems like sol-gels, which could otherwise in theory be used in a similar context. Regarding the issue of novelty, we do not agree with the reviewer that the simple fact we use size exclusion means the idea is not novel. Instead, we would argue that our application of size exclusion is quite innovative, using an old principle to solve what has been a long-standing problem in the field of drug screening. We believe the significance of this result, coupled with the novel way we have printed our hydrogels (using reactive inks and solenoid printing, approaches amenable to high-throughput production), makes this work both highly novel and highly impactful.

Reviewer #2 (Remarks to the Author): The authors have carefully addressed my critique about high enzyme concentration, and I am satisfied with their response. I would note that it is now clear that they do lose signal to noise in this format--as for plate-based aqueous screens, for instance, far less TEM-1 can be used--and I think that this caveat could be strengthened even slightly further, as it may affect all enzymes. Still, this is a quibble and I certainly support publication of this manuscript. Their responses to the other reviewers also seemed thoughtful and represent substantial work, though naturally I will defer to the other reviewers here. Still, certainly from my standpoint, the manuscript can be published. I think it will find an interested readership.

Response: We thank the reviewer for the positive comments. Regarding the issue of signal:noise, we do not have definitive data one way or another showing a clear difference between the immobilized and solution assays in terms of sensitivity; all the enzymes we tested have IC50 values that are well within the accessible signal:noise range of both assays. The reviewer is however correct that the lowest detectable concentration using our printed assay (~5 nM) is slightly higher than that achievable (at least of other β -lactamase isoforms) in a solution assay (~1 nM or slightly lower). This is certainly not unusual for an immobilized enzyme assay and still represents a very low threshold of detection. Furthermore, the substantially lower volume required for the printed assay results in similar or lower total protein amounts being required for performing the printed assay versus the solution assay. Thus, even if higher concentrations of enzymes are required in the printed assay, the total amount of enzyme required is actually similar or less. However, to be sure we do not over-

generalize the results for different enzymes, we have specified in the manuscript that the lower limit of detection we cite is just for β -lactamase.

Reviewer #3 (Remarks to the Author): Q1. The authors have made great efforts to persuade reviewers. I agree with many of the parts described by the author. However, as reviewer 1 point out, quantitative comparison with other hydrogel systems than solution assays will make this manuscript more strong.

Response: We appreciate the reviewer's comment here. In principle, we agree that quantitative comparisons with other hydrogel or sol-gel systems would be interesting; indeed, if the manuscript was focused only on a new enzyme immobilization method using a mixing-induced hydrogel, we would agree with the reviewer that such a comparison would be important to show. However, given that main conclusions focus around the unique performance properties of the immobilized enzyme system in terms of differentiating between true and non-specific inhibitors, we do not feel that reproducing this same performance with other hydrogel systems is important to support to the main conclusions given in this manuscript; indeed, it would in our opinion distract from the main message of the paper. In addition, even if such an experiment was deemed important, as we noted in our previous response there are significant feasibility issues in terms of applying other types of gelation chemistries to reproduce our current test system. The mixing-induced printing process used to create the microzones of the high-throughput assay test plate has unique advantages (as outlined in our earlier response to reviewer #1) which make recreating this assay in a comparable manner with other chemistries very nontrivial. As such, any comparison made would be somewhat indirect in nature, decreasing its value. Q2. Although UV polymerization is cytotoxic and poor, many studies have already encapsulated and cultured cells (even stem cells) with photocrosslinking. The UV polymerization system mentioned by the authors seems to be an example of the worst case. It would be wrong to mention that photocrosslinking is unconditionally bad. Response: While we stand by the described benefits of our gelation system versus UV photopolymerization, we also appreciate the reviewer's concern that we are over-emphasizing the drawbacks of UV photopolymerization, as indeed it has been used successfully for stem cell encapsulation among many other applications. As such, we have rephrased the drawbacks of UV polymerization stated in the manuscript to ensure it is clear UV photopolymerization is not unconditionally bad as well as emphasize the benefits associated with our avoidance of a secondary processing step (light irradiation), regardless of the potential cytotoxicity of that secondary processing step, using the mixing-induced hydrogel printing approach: This method allows for a covalently crosslinked hydrogel to be printed in the absence of UV photopolymerization to facilitate gelation, avoiding the need for a secondary processing step as well as eliminating the potential for enzyme denaturation sometimes observed due to the radical species generated during polymerization.